# Exploring the sustainability of virtual care interventions: A scoping review

Tujuanna Austin[1]*, Farah Tahsin[1], Darren Larsen[1,2], Ross Baker[1], Carolyn Steele Gray[1,3]

1 Institute of Health Policy, Management and Evaluation, University of Toronto, Toronto, Canada,
2 Women's College Hospital, Toronto, Canada, 3 Bridgepoint Collaboratory Lunenfeld-Tanenbaum Research Institute Sinai Health System, Toronto, Canada

* Tujuanna.austin@mail.utoronto.ca

## Abstract

During the COVID-19 pandemic, virtual care has proven instrumental in ensuring the continuity of healthcare services. In the context of virtual care's growing prominence and continued use, understanding how and why virtual care interventions are sustained will help healthcare systems to better prepare for future crises. The objectives of this scoping review were to construct a conceptualization of of virtual care sustainability and to describe factors influencing the sustainability of virtual care, shedding light on the determinants that shape its longevity and continued use. Literature describing the sustainability of virtual care interventions was summarized. Details of the intervention, setting, methodology, description and evidence of sustainability, and synopsis of key findings were documented. The charted data were summarized to gain a descriptive understanding of the data collected and to establish patterns. A conceptualization of virtual care intervention sustainability focused on the concepts of fidelity and adaptability. Sustainability of virtual care interventions were conceptualized as the intervention's ability to continue to be used according to its initial design, the extent to which the intervention continued to achieve its intended outcomes (fidelity), and the ability of the intervention to evolve as the context in which it is used also evolves (adaptability). While there were various definitions of sustainability referenced, no included studies mentioned a definition of sustainability specific to virtual care. Commonalities in definitions included the continued use of virtual care and the continuation of the benefits of virtual care for some period of time. Findings indicate that there is no "one size fits all" approach to achieving sustainability of virtual care interventions, but instead identify factors that may support or hinder sustainability. Important to understanding sustainability of virtual care interventions, is the complexity of the interactions that influence it. Specifically, the factors of fidelity and adaptability are found to be important to understanding the sustainability of virtual care interventions.

which permits unrestricted use, distribution, and reproduction in any medium, provided the original author and source are credited.

**Data availability statement:** The data that support the findings of this study are included in the manuscript.

**Funding:** The author(s) received no specific funding for this work.

**Competing interests:** The authors have declared that no competing interests exist.

## Author summary

This research highlights the critical role of virtual care interventions and explores how these interventions can be sustained in the long term. By conducting a scoping review, the study aims to develop a conceptualization (i.e., representation of key concepts and their relationships) for understanding the sustainability of virtual care, examining key factors that influence its continued use and effectiveness. The findings suggest that sustainability is not a one-size-fits-all scenario; instead, it involves a complex interplay of factors that can either promote or hinder the ongoing success of virtual care interventions. The study emphasizes the importance of two main aspects: fidelity, which refers to how closely the intervention adheres to its original design, and adaptability, or how well the intervention can evolve with changing circumstances. This research is significant as it provides valuable insights for healthcare systems aiming to prepare for future challenges, ensuring that virtual care can continue to deliver its benefits. By identifying the determinants of sustainability, this work paves the way for more effective strategies to integrate virtual care into everyday healthcare practices, ultimately improving patient outcomes and system resilience.

## Introduction

In recent years, the integration of virtual care into healthcare systems has revolutionized the delivery of medical services, offering unprecedented accessibility, efficiency, and flexibility [1,2]. Virtual care is defined as "any interaction between patients and/or members of their circle of care, occurring remotely, using any forms of communication or information technologies to facilitate or maximize the quality and effectiveness of patient care" [3]. Virtual care has been shown to reduce unplanned and avoidable admissions to hospitals by following patients more closely in their own homes [4]. Virtual care can enable high-quality access to care that ultimately enhances provider-patient interactions while presenting opportunities to inform, personalize, accelerate, and augment care across the care delivery spectrum, from disease prevention to treatment to ongoing monitoring [5]. Moreover, these systems have the potential to provide primary care practitioners with more efficient ways to manage patient care and work in multidisciplinary teams [6].

The COVID-19 pandemic created a global health crisis and necessitated innovative solutions to sustain healthcare delivery amidst physical distancing measures and overwhelming demands on healthcare infrastructure [7–10]. Virtual care, encompassing telemedicine, remote monitoring, and digital interventions, has emerged as a critical tool in the healthcare arsenal. The pandemic has thrust virtual care initiatives to the forefront as practitioners shifted to delivering care remotely [8,11]. The Canadian Institute for Health Information reported that in the first year of the pandemic (April 2020 to March 2021), virtual care visits averaged 244 services per 1,000 people per month, compared with an average of 52 the year before [12]. In the United States, virtual care encounters increased 766% in the first 3 months of the pandemic

[13]. Studies conducted to assess the impact of the rapid shift to virtual care showed that patients report satisfaction with virtual care and up to one-third would like virtual care to be the first point of contact following the pandemic [14–16]. From providers' perspectives, 70% of healthcare providers believe virtual care improves patient access and enables quality care and efficient care for their patients and 64% intend to maintain or increase its use following the pandemic [17]. Sustained use of virtual care may continue to be important as a strategy to address the a rise in chronic conditions, a potential echo pandemic due to mental health issues, and an aging population [15,18,19].

The COVID-19 pandemic served as a driver for the adoption, expansion and innovation of virtual care [20–24]. Tele-medicine, remote monitoring, and digital health platforms, which had previously faced slow uptake due to regulatory, financial, and cultural barriers, were suddenly deployed at scale [25,26]. The pandemic created a natural experiment in digital health innovation, offering the opportunity to investigate virtual care interventions are most effective, for which populations, and under what conditions [27].

While virtual care has many benefits, it is not without its shortcomings. The disadvantages of virtual care include the inability to perform physical examinations and procedures, difficulty establishing therapeutic relationships, managing complex mental health issues, and the lack of comfort that is inherent to in-person visits [28]. Privacy, continuity of care, and equity of access are additional challenges associated with virtual care [28,29]. Concerning equity, virtual care requires access to the required technology and the ability to use that technology, which may be a barrier to certain patient populations [4]. Furthermore, virtual care may be inappropriate for some encounters such as sharing an end-of-life diagnosis with a patient or conducting certain mental and physical health assessments [30].

Many professional associations, including the Canadian Medical Association and American Medical Association embrace virtual care as a means of health service delivery [13,31]. These associations acknowledge that patient demand and virtual care's ability to improve access mean that it will become increasingly prevalent. Despite these endorsements, there is the danger that virtual care will remain fragmented and be delivered inequitably without concrete steps to embed virtual care in healthcare systems [3,17,31,32]. Even as virtual care emerged as a key and critical mechanism for delivering healthcare in response to the COVID-19 pandemic, there is a long way to go in terms of understanding the potential and opportunities surrounding the sustained use of technologies to deliver healthcare [31]. In a rapidly changing environment such as this, it is important to explore the sustainability of virtual care to build a conceptual basis for understanding resilience and utility of interventions in healthcare settings.

As we embrace the digital era via the increasing reliance on and use of virtual care, its sustainability emerges as a critical concern, demanding in-depth exploration. Currently the sustainability of virtual care interventions is not uniformly defined in the literature, which has hindered the development of a comprehensive conceptual framework and evaluation of the topic. There is consensus that conceptualizing sustainable interventions as static tools fails to consider the complex adaptive nature of health care systems (7, 11, 28, 30). Based on the literature, it is known that an innovation may be sustained in one setting, but be abandoned in another; moreover, the conceptualization of sustainability in one setting may not be reflected in another [33–37]. Some research focuses on sustainability of effects of an innovation while others focus on sustainability of the innovation (or parts of the innovation) itself [37–43]. There is considerable debate in the literature as to what constitutes a 'sustained' initiative; some argue that it must be measured in terms of fidelity to the intervention as originally designed, while others draw attention to the process by which innovations naturally evolve through time, giving rise to inevitable 'deviation' from the intervention [44,45]. Implementation science literature recognizes that sustained healthcare innovations require a balance of standardization and responsiveness to local contexts for long-term viable to occur [27,38]. For instance, frameworks like the Dynamic Sustainability Framework emphasize that rigid adherence to protocols often conflicts with real-world needs for flexibility [38]. It is therefore important to explore sustainability as related to virtual care so as to develop a better understanding of how virtual interventions can be sustained in healthcare settings.

The objectives of this paper are twofold: first, to construct a conceptualization of virtual care sustainability building on extant literature, emphasizing its continued evolution and incorporation into healthcare environments. Second, to describe

factors influencing the sustainability of virtual care, shedding light on the determinants that shape its longevity and continued use.

## Materials and methods

The methodology for this scoping review is guided by the framework by Arksey and O'Malley (2005) [46]. Arksey and O'Malley outline five steps for conducting a scoping review: identifying the research question, identifying relevant studies, study selection, charting the data, and collating, summarizing, and reporting the results.

### Identifying the research question

In this study, the research question investigated is, "what is known in the literature about the sustainability of virtual care interventions in health service delivery settings?" The research objectives of this review were to construct a conceptualization of framework of virtual care sustainability and to describe factors influencing the sustainability of virtual care.

### Identifying relevant studies

The search strategy of the scoping review was informed by search terms and databases used in other literature syntheses focusing on sustainability [35,38,39,41,44,47–50] and the selection of relevant key terms and relevant Medical Subject Headings (MESH) from included databases. Search terms included, "sustainability", "virtual care", "health care", and their conjugates and synonyms. Boolean operators were used to account for plurals and variations in spellings. The MEDLINE, Embase, CINHAL, Scopus, and Cochrane databases were searched; these databases span the health sciences, allied health, and multidisciplinary literature. Three searches were conducted: the initial search in August 2022, a second search in April 2023, and a third in May 2024 (using the same search terms) to capture any recent relevant literature.

### Study selection

Studies were included if they were published in English and addressed the sustainability of a virtual care intervention in a healthcare setting. Studies were excluded if they were not published in English, and did not present a definition, framework, or reference to adefinition or conceptualization (including contributing factors to) of the term sustainability. For this study, in line with the definition of virtual care, a virtual care intervention encompasses any intervention that uses technology to improve or deliver patient care where the patient and their care team are not in the same location (e.g., video consultations, telephone, mobile applications, remote monitoring). Studies that do not discuss a specific virtual intervention or studies where sustainability was not a specific concern of the study (e.g., the study focused only on the adoption and initial implementation of the intervention) were excluded. Table 1 presents the inclusion and exclusion criteria.

Studies were screened at the title/abstract level, then again at the full-text level using the online screening tool Covidence [51]. Duplicate independent screening of titles and abstracts categorized papers into the categories relevant, irrelevant, or unsure by two researchers (TA, FT). Any papers in the unsure category were discussed between the two

**Table 1. Inclusion and exclusion criteria for the scoping review.**

| Inclusion Criteria | Exclusion Criteria |
|---|---|
| • Paper discusses the concept of sustainability | ✗ Paper did not present a definition, framework, or reference to the authors' selected definition of the term sustainability |
| • Paper examines includes a virtual care intervention | ✗ Paper did not discuss a specific virtual intervention |
| • Setting of a paper is in a healthcare environment | ✗ Intervention did not apply to the healthcare setting |
| | ✗ Paper focused only on adoption and initial implementation of the intervention |

**Table 2. Summary of data collation.**

| Data Collation Category | Description |
|---|---|
| Definition of sustainability | How did the authors describe sustainability? What concepts did the authors include their definition of sustainability (e.g., length of time)? |
| Purpose of virtual care intervention | What was the virtual care intervention intended to do (e.g., remote patient monitoring)? |
| Functionalities of virtual care intervention | What were the features of the virtual care intervention (e.g., ability for patients to input health data into a mobile application)? |

reviewers until consensus was met. At the full-text level, duplicate independent screening was conducted for the first 10% of all studies retrieved from (n = 660). A threshold of a 90% kappa score (agreement between the reviewers) indicated that the independent screening could continue. The two reviewers (TA and FS) achieved a kappa score of 93% and the remaining 90% of full texts were screened independently.

### Charting the data

Using Microsoft Excel, data from the studies were extracted by TA into the following categories: details of the virtual care intervention (i.e., information on the purpose, objectives, functionalities, modifications, and extent of use); details about the study population, setting, participant demographics, study design and methods used; study setting; authors description of sustainability; evidence of sustained change (e.g., length of time that the intervention was delivered, any associations reported by the authors about intervention and sustained effectiveness); and a synopsis of key findings. Through an iterative coding process, TA and FT identified and categorized emerging factors influencing sustainability, which were then grouped into broader themes (e.g., system-level enablers).

### Collating, summarizing, and reporting the results

The charted data were summarized (refer to Table 2) to gain a descriptive understanding of the data collected and to establish patterns. Summaries focused on how sustainability was defined, how the virtual care intervention was sustained, and contributing factors to intervention sustainability. From these summaries, themes (based on the patterns in the summaries) were developed. The themes were: 1) commonalities in definitions of sustainability; 2) evolution of the intervention over time (adaptability); and 3) fidelity of the intervention to its original design. Commonalities in the definitions of sustainability and the purpose and functionalities of the virtual care intervention were also assessed and differences between the setting in which the virtual care intervention was used were noted and assessed in terms of contributing factors for sustainability.

## Results

Database and grey literature search results from both searches returned 7476 papers, following duplicate removal. After title/abstract and full text screening, 25 studies, detailed in Table 3, were eligible for inclusion. Fig 1 (below) presents a flow chart of studies excluded at various stages of the screening process.

Of the 25 papers included in this scoping review, 9 studies investigated video consulting or conferencing interventions [16,52,56,59,68,72–75], 1 investigated virtual medication management intervention [53], 5 investigated telephone interventions [57,61,62,65,66], 6 investigated telephone and video interventions [63,64,67,69–71], and 4 investigated various synchronous and asynchronous interventions [54,55,58,60]. Settings ranged from hospital [72,73,75], mental health [52,53], orthopedics [57], palliative care [71], chronic disease management [70], pediatric care [58], primary care [16], cardiac care [67,69], homecare [61,62,64,74], neurology [63], community care [54], health system [59,60,65], and studies investigating various settings [55,56,68].The following section structures the results as they pertain to defining sustainability of virtual care interventions (with a particular focus on the notions of adaptability and fidelity) and other factors important to conceptualizing the topic.

**Table 3. Overview of included studies.**

| Study Title (Author, Date) | Virtual Care Intervention | Study Objectives | Study Population/ Setting | Methods | Operationalization of Sustainability | Evidence of Sustainability | Synopsis of Key Findings |
|---|---|---|---|---|---|---|---|
| Implementing and Sustaining Team-Based Telecare for Bipolar Disorder: Lessons Learned from a Model-Guided, Mixed Methods Analysis [Bauer et al., 2018] [52] | Clinical video conferencing (bipolar telehealth program) | To characterize the extent of implementation and sustainability of the program after its establishment and to identify barriers and facilitators to implementation and sustainability | Mental health | Mixed methods program evaluation: assessment of the quantitative aspect of the RE-AIM framework and semi-structured interviews with providers using the intervention | Sustainment of the intervention beyond the initial consultations for at least 2 years | Statistically significant growth in use of video consultations and sites using the intervention beyond the first year | Facilitators: valued recommendations, ease of use and integration into ongoing workflow via electronic health record for consulting providers, and extensive infrastructure at the national level to support implementation. Barriers: labor-intensive nature of scheduling; variable availability of telehealth space, equipment, and staff at certain sites |
| Walk-In Telemental Health Clinics Improve Access and Efficiency: A 2-Year Follow-Up Analysis [Neufeld and Case, 2013] [53] | Telemedicine services (virtual medication evaluations and medication management) | To evaluate changes in access, quality of care and service efficiency | Multisite rural community mental health centre | Quantitative evaluation of access, quality, outcomes and costs of the program (baseline and 24-month comparison) | Ability of clinicians to maintain the same level of quality while seeing as many patients as possible during the times they were scheduled at each site | Improvement in access, quality, and service efficiency compared with similar services delivered using traditional methods (20-percentage-point advantage in conversion of scheduled time to billable time over traditional clinics) | A large proportion of improvements were attributable to the 20% of clinic volume that was open-scheduled or "walk-in" in nature. |
| Multiple pathways to scaling up and sustainability: an exploration of digital health solutions in South Africa [Swartz et al., 2021] [54] | 4 digital health programs: two mortality audit reporting and visualization tools; one patient messaging and feedback service and a data capture and decision-support application | To explore digital health solutions that have or are anticipated to reach national scale in South Africa | Community health setting | Qualitative case study design | Ability of a digital solution for health to persist long term | Continued use of the programs after adapting the programs to local contexts | Factors contributing to scaling and sustaining digital health: technology, actors, value proposition, context. The scale and continued use were driven by the use of champions, leadership value recognition, simplified technology government buy-in and ongoing financial and technical support from donors and technical partners. |

(Continued)

Table 3. (Continued)

| Study Title (Author, Date) | Virtual Care Intervention | Study Objectives | Study Population/ Setting | Methods | Operation-alization of Sustainability | Evidence of Sustainability | Synopsis of Key Findings |
|---|---|---|---|---|---|---|---|
| Keys to a success-ful and sustainable telemedicine program [Whitten & Nguyen, 2010] [55] | Various telemedicine programs | To determine organizational characteristics evident in successful telemedicine programs | Various health-care settings | Online survey | Continuation of telemedicine program | Continued and ongoing use of the interventions | To have a sustainable telemedicine program, it is important to define the program as its own entity with a formal structure, designed with the objective to improve quality of patient care, and have financial stability |
| Advantages and limitations of virtual online consultations in a NHS acute trust: the VOCAL mixed-methods study [Shaw et al., 2018] [56] | Remote video consultations | To explore the social and technical factors that support video consultations, in order identify the sustainability of this service model in areas where it proves to be acceptable and effective | Diabetes, ante-natal diabetes, and cancer surgery | Multilevel mixed-methods study | Sustained benefits, sustained use of and sustained engagement with video consults | 2% and 20% of all consultations were being undertaken remotely in participating clinics | Embedding video consultations in existing workflows on an ongoing basis was timely and resource intensive. When clinical, technical and practical preconditions were met, virtual consultations appeared to be safe and were popular with both patients and staff |
| Virtual phone clinics in orthopaedics: evaluation of clinical application and sustainability [Pradhan et al., 2021] [57] | Telephone visits | To determine how to use virtual orthopedic clinics effectively and sustainably by comparing virtual and in-person clinics in terms of patient and clinician satisfaction in orthopedic consultations and evaluating the costs associated with virtual care | Orthopedics | Survey | Patient satisfaction | Higher patient satisfactions scores were associated with appointments for delivering results and where patients felt clinical examination was not needed | High mean patient satisfaction score, however, significantly lower when compared to in-person; patients being managed non-operatively and being followed up after surgical intervention reported higher satisfaction scores for in-person vs. virtual care. Current virtual care structures use greater clinician resources and generates lower reimbursement (11.5% lower) than in-person |

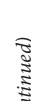

**Table 3.** (Continued)

| Study Title (Author, Date) | Virtual Care Intervention | Study Objectives | Study Population/ Setting | Methods | Operationalization of Sustainability | Evidence of Sustainability | Synopsis of Key Findings |
|---|---|---|---|---|---|---|---|
| Innovative virtual care delivery in a Canadian paediatric tertiary-care centre [Goldbloom et al. 2022] [58] | Various virtual care interventions (video conferencing, virtual family-centred rounds, virtual emergency room) | To highlight the transferable unique ways the hospital has integrated virtual care delivery with intention of sustainability | Various departments in a pediatric hospital | Interviews | Continuation of high volume and high quality pediatric care | Continuation of video conferencing and adaptation of each model to the patient population | Innovation and creativity drove the development of clinically diverse virtual care programs. Facilitators: virtual care leadership, development of new workflows, multi-disciplinary meetings, ongoing redesigning |
| Exploring the use and challenges of implementing virtual visits during COVID-19 in primary care and lessons for sustained use [Mohammed et al., 2021] [16] | Synchronous and asynchonous virtual care (Video visits, securing messaging, phone calls, text messaging) | To assess primary care practitioners' use of virtual visits in primary care settings across Southwestern Ontario after the first wave of the COVID-19 pandemic. | Primary care | Descriptive cross-sectional study | Use of virtual care beyond the first wave of the COVID-19 pandemic | Continued use of virtual care | Challenges to sustainability: patient access to technology, patient knowledge of technology, connectivity issues, lack of integration with electronic medical record |
| Achieving Spread, Scale Up and Sustainability of Video Consulting Services During the COVID-19 Pandemic? Findings From a Comparative Case Study of Policy Implementation in England, Wales, Scotland and Northern Ireland [Shaw et al., 2021] [59] | Video consulting | To unpack what could or should happen to achieve spread and scale up of video consulting services during a time of crisis | National Health Service | Comparative case study | Spread and scale-up of virtual care during the pandemic | Evolution of and continued use of video consulting | Facilitators: pre-existing support for virtual care, reduced regulation, simplified procurement processes, and efforts at system level to develop, review and run video consulting programs. Barriers: limited infrastructure |
| Barriers and facilitators for the sustainability of digital health interventions in low and middle-income countries: A systematic review (Kabore et al., 2022] [60] | Information and communication technologies designed to improve health systems and services | To identify the barriers and facilitators for the sustainability of digital health interventions in low and middle-income countries | Low and middle income countries | Systematic review | Sustainability as a dynamic process, and that goals and strategies for achieving it must continuously adapt to changing environmental conditions | Continuous manifestation of benefits and outcomes of digital innovations for health workers, the standard of healthcare, and patient experience | Facilitators: strong commitment and involvement of relevant stakeholders, experience and confidence in using the system, motivation and competence of staff. Barriers: infra-structure, equipment, internet, electricity and technology |

*(Continued)*

| Study Title (Author, Date) | Virtual Care Intervention | Study Objectives | Study Population/ Setting | Methods | Operationalization of Sustainability | Evidence of Sustainability | Synopsis of Key Findings |
|---|---|---|---|---|---|---|---|
| Unsustainable Home Telehealth: A Texas Qualitative Study [Radhakrishnan et al., 2016] [61] | Home telehealth | To explore the reasons for the initial adoption and the eventual decline of a decade-long home telehealth program at a Texas home health agency | Homecare | Qualitative study | Ongoing program use | Lack of significant impact on patient outcomes, and financial, technical, management, and communication-related challenges | Themes related to the home telehealth program's decline: impact on patient-centered outcomes, impact on cost-effectiveness, communication and collaboration, technology usability, and home health management culture |
| Transitioning a home telehealth project into a sustainable, large-scale service: a qualitative study [Wade et al., 2016] [62] | Various telehealth projects | To produce a sustained approach for adding home telehealth to existing rehabilitation, palliative and geriatric outreach services | Homecare for rehabilitation, palliative and geriatric programs | Qualitative study | The ability to continue operating into the future without obvious threats | Eventual expansion of telehealth to rehabilitation, palliative and geriatric programs | Key factor for large-scale sustainability is leadership support, which is enabled by showing solutions to the problems, demonstrating how home telehealth aligns with health service policies, and achieving clinician acceptance |
| Sustaining telecare consultations in nurse-led clinics: Perceptions of stroke patients and advanced practice nurses: A qualitative study [Wong et al, 2023] [63] | Telephone and video visits | To ascertain the experiences of stroke survivors and healthcare providers regarding the utilization of a post-stroke telecare service in Hong Kong | Neurological medical ward of hospital | Qualitative study | Continued/long-term use | Continued use of the program after adapting to the COVID-19 context | Perception of telecare program as a significant alternative and one that complements conventional face-to-face follow-ups |
| Barriers and Facilitators for Sustainability of Tele-Homecare Programs: A Systematic Review [Radhakrishnan et al., 2016] [64] | Telehomecare: communication and clinical information system that enables the interaction of voice, video, and health-related data using telephone lines from the patients' homes to their home health agencies in conjunction with nurses' home visits | To identify the barriers and facilitators for sustainability of tele-homecare programs implemented by home health nursing agencies for chronic disease management | Homecare | Systematic review | The use of tele-homecare services that hold the promise of being absorbed into routine health care delivery | Ongoing use of telehomecare | Barriers to and facilitators for sustainability: perceptions of effectiveness, tailoring to patients, nurse–patient communication, interprofessional collaboration, organization of process and culture, and technology |

*(Continued)*

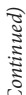

| Study Title (Author, Date) | Virtual Care Intervention | Study Objectives | Study Population/ Setting | Methods | Operation-alization of Sustainability | Evidence of Sustainability | Synopsis of Key Findings |
|---|---|---|---|---|---|---|---|
| Success factors for telehealth—A case study [Moehr et al., 2006] [65] | Two clinical application telehealth services | To present the lessons learned from an evaluation of a comprehensive telehealth project regarding success factors and evaluation methodology | Province-wide telehealth infrastructure | Program evaluation | Continued use | One clinical application domain was cancelled after 6 months, the other continues | Facilitators: focus on chronic conditions which require visual information for proper management and involvement of established teams in regular scheduled visits. Barriers: lack of time for preparation and establishment of routine use |
| Beyond forced telehealth adoption: A framework to sustain telehealth among allied health services [Thomas et al., 2022] [66] | Telehealth | To determine the clinician, service, and system level factors that influence sustained use of telehealth and develop a framework to enhance sustained use where appropriate | Metropolitan health service network consisting of four hospitals | Multi-method observational study | Sustained used of telehealth | Increased telehealth use during the peak COVID period reverted to in-person activity as restrictions eased | Forced telehealth adoption has increased clinician reluctance; Value proposition for clinicians is lacking; Lack of organizational readiness inhibited telehealth use - hybrid care needs to be integrated; Clinicians perceive limited consumer demand for telehealth - greater consumer-end support required |
| Expanding Telemonitoring in a Virtual World: A Case Study of the Expansion of a Heart Failure Telemonitoring Program During the COVID-19 Pandemic [Wali et al., 2021] [67] | Telemonitoring | To understand the experiences related to the expanded role of a telemonitoring program under the changing conditions of the pandemic | Hospital | Case study | Ongoing program use and ongoing benefits | Adaptation of the program to the COVID-19 context | Four themes were identified: providing care continuity through telemonitoring; adapting telemonitoring operations for a more virtual health care system; confronting virtual workflow challenges; and fostering a meaningful patient-provider relationship. Barriers to sustainability: lack of system integration and alert-driven interactions |

*(Continued)*

Table 3. (Continued)

| Study Title (Author, Date) | Virtual Care Intervention | Study Objectives | Study Population/ Setting | Methods | Operation-alization of Sustainability | Evidence of Sustainability | Synopsis of Key Findings |
|---|---|---|---|---|---|---|---|
| Spread, Scale-up, and Sustainability of Video Consulting in Health Care: Systematic Review and Synthesis Guided by the NASSS Framework [James et al., 2021] [68] | Video consulting | To review and synthesize opportunities, challenges, and lessons learned in the scale-up, spread, and sustainability of video consultations, and to identify transferable insights that can inform policy and practice | Various including rehabilitation, geriatrics, cancer surgery, diabetes, mental health, and primary care | Systematic review | 7 domains of the NASSS framework | Applicability of the NASSS framework to the video consulting intervention | Enablers: embedded leadership and the presence of a telehealth champion, appropriate reimbursement mechanisms, user-friendly technology, pre-existing staff relationships, and adaptation over time. Challenges: absence of a long-term strategic plan, resistance to change, cost/reimbursement issues, and technical experience of staff |
| User-Centered Adaptation of an Existing Heart Failure Telemonitoring Program to Ensure Sustainability and Scalability: Qualitative Study [Ware et al., 2018] [69] | Telemonitoring | To understand which components of the telemonitoring program could be modified to reduce costs and adapted to other contexts while maintaining program fidelity and to describe the changes made to the telemonitoring program to enable its sustainability within the initial implementation site and scalability to other health organizations | Specialty heart failure clinic | Qualitative study | Adaptability of the program to other settings | Changes to the program to facilitate ongoing use | Opportunities for cost reduction and adaptability: Bring Your Own Device, technical support, clinician role, duration of enrollment, and intensity of monitoring. |
| The Time Is Now: A Guide to Sustainable Telemedicine During COVID-19 and Beyond [Shah et al., 2020] [70] | Telemedicine: the application of information and communication technologies for providing healthcare services at a distance without the need for direct contact with the patient | To outline important considerations on telemedicine | Chronic disease management | Qualitative study | Integration of telemedicine with existing workflow | Use of telemedicine during the COVID-19 pandemic | Requirements for sustainability: reimbursement, insurance, appropriate technology, privacy and guidance on appropriate use |

(Continued)

| Study Title (Author, Date) | Virtual Care Intervention | Study Objectives | Study Population/ Setting | Methods | Operationalization of Sustainability | Evidence of Sustainability | Synopsis of Key Findings |
|---|---|---|---|---|---|---|---|
| "Never waste a good crisis". A qualitative study of the impact of COVID-19 on palliative care in seven hospitals using the Dynamic Sustainability Framework [Holdsworth et al., 2022] [71] | Virtual palliative care programs | To understand how the pandemic impacted the implementation of new and existing palliative care programs in diverse hospital systems using the Dynamic Sustainability Framework | Palliative care | Qualitative study | Adaptation of existing programs to virtual modalities | Continued use of virtual programming and implementation of new virtual programming | Stresses caused by the COVID-19 pandemic did not impact palliative care programs equally; inpatient programs, whether new or existing, had greater success in adapting to the pandemic environment than new outpatient services and community-based services which were hampered by lack of in-person contact necessary for building relationships to establish a new service |
| Factors influencing the sustainability of digital health interventions in low-resource settings: Lessons from five countries [McCool et al., 2020] [72] | mHealth platforms, video consultations, phone follow-ups | To identify key factors enabling or hindering the long-term sustainability of digital health interventions in low-resource settings | Rural primary and secondary care facilities | Qualitative study | Continued use and institutionalization of digital health interventions beyond pilot phases, with maintained benefits for health systems and users | Continued use and institutionalization of telemedicine and mHealth interventions | Enablers of sustainability: integration into national policies, co-design of intervention, adaptive tools Threats to sustainability: short-term funding, mismatch between technology and local digital literacy, no evaluation |
| Facilitating telemedicine project sustainability in medically underserved areas: a healthcare provider participant perspective [Paul et al., 2016] [73] | Video consultations and remote patient monitoring | To explore healthcare providers' perspective on factors influencing the sustainability of telemedicine projects in medically underserved areas | Small clinics and community hospitals | Qualitative study | Ongoing routine use of telemedicine services integrated into standard care practices, supported by stable resources and stakeholder commitment | Continued use of video consultations and remote monitoring projects | Success factors of sustainability: strong leadership buy-in, provider training, Medicaid coverage for virtual visits Barriers to sustainability: technical issues, workflow disruptions |

*(Continued)*

**Table 3.** (Continued)

| Study Title (Author, Date) | Virtual Care Intervention | Study Objectives | Study Population/ Setting | Methods | Operationalization of Sustainability | Evidence of Sustainability | Synopsis of Key Findings |
|---|---|---|---|---|---|---|---|
| Exploring factors that affect the uptake and sustainability of videoconferencing for health-care provision for older adults in care homes: a realist evaluation [Newbould et al., 2021] [74] | Videoconferencing for remote consultations | To identify the usability and sustainability of implementing video-conferencing in care homes | Residential care home | Realist evaluation | Continued and meaningful use of videoconferencing as an integrated component of routine care | Establishment and continued use of regular virtual clinics | Context: strongest adoption and sustainability occurred in homes with existing technology infrastructure, management commitment, established relationship with healthcare providers Mechanism: normalization, workflow integration, perceived benefits |
| Sustaining e-health innovations in a complex hospital environment: learning through evidence [Jaana et al., 2024] [75] | Remote patient monitoring, mHealth applications, virtual wound care consultations | To identify critical success factors for sustaining e-health innovations | Large, tertiary care hospital | Mixed-methods case study | The continued and effective use of e-health innovations beyond initial implementation, where the technology becomes a part of routine clinical practice and delivers ongoing value | Remote patient monitoring system becoming standard of care, >90% clinician adoption rate | Organization learning capacity, clinical relevance, and governance structures are important for sustainability |

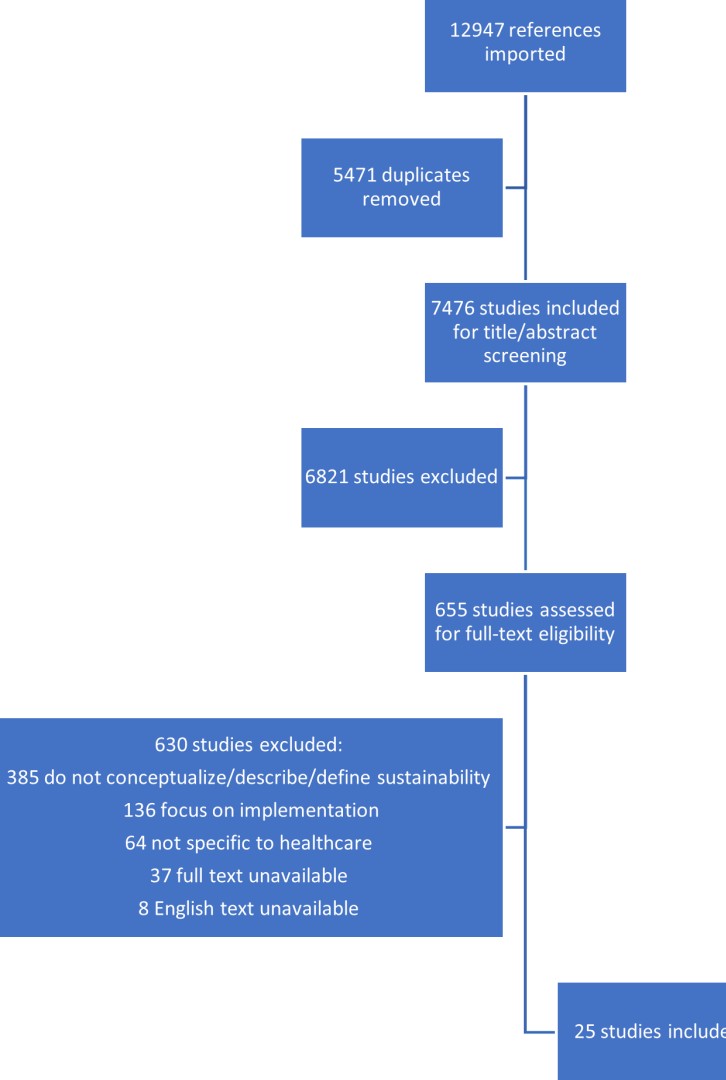

**Fig 1. Flowchart of included and excluded studies.**

## Definitions of sustainability

Among the articles included in this scoping review, there were various interpretations of sustainability. Papers referenced a variety of definitions of sustainability of healthcare interventions, however, no studies included in this review mentioned a definition of sustainability specific to virtual care. While some studies cited other definitions in their entirety [52,68,75,76], others referred to elements of those definitions in their interpretations of sustainability [53–55,74]. Table 4, below, provides a synopsis of the sustainability definitions of healthcare interventions referenced by papers included in this scoping review. While some definitions were more detailed than others, commonalities amongst definitions included the continued use of virtual care and the continuation of the benefits of virtual care for some period of time.

When describing *how* virtual care interventions were able to be continuously used and/or how the benefits were able to be continuously realized, fidelity of the virtual care intervention to its initial design; and adaptability of the virtual care intervention to the context in which it's used were frequently mentioned.

**Table 4. Sustainability definitions referenced in papers included in the scoping review.**

| Definition | Source of Definition | Papers Referencing the Definition |
|---|---|---|
| Shediac-Rizkallah and Bone's framework for conceptualising programme sustainability considers project design and implementation factors, factors within the organisational setting, and factors in the broader community environment as contributors to sustainability | Shediac-Rizkallah and Bone (1998) [77] | [60] |
| Non-adoption, Abandonment and Challenges to the Scale-up, Spread and Sustainability of Health and Care Technologies (NASSS) Framework: This framework defines scale-up, spread and sustainability as it relates to 7 domains: health condition, technology, value proposition, adopter system, organisation, wider system and adaptation over time | Greenhalgh et al., (2017) [33] | [59,66,68,73,76] |
| Reach, Efficacy, Adoption, Implementation, and Maintenance (RE-AIM) Framework: Maintenance is operationalized at the individual level (e.g., long-term effectiveness or impact) and the setting level (e.g., sustainability program components after original implementation) | Glasgow et al., 2019 [78] | [52] |
| Sustainability is defined as a function of: (1) whether, and to what extent, the core elements (the elements most closely associated with desired health benefits) are maintained; (2) the extent to which desired health benefits are maintained or improved upon over time after initial funding or supports have been withdrawn; (3) the extent, nature, and impact of modifications to the core and adaptable/peripheral elements of the program or innovation and (4) continued capacity to function at the required level to maintain the desired benefits | Stirman et al., 2012 [50] | [72,75,79] |
| Sustainability is described as the ability of clinicians to maintain the same level of quality while seeing as many patients as possible | Neufeld et al., 2013 [53] | [53] |
| Sustainability is described as the ability of a digital solution for health to persist long term | Swart et al., (2021) [54] | [54] |
| Sustainability is defined as the continuation of a telemedicine program for a long period of time | Whitten and Ngyen, 2010 [55] | [55] |
| Sustainability is described as a function of sustained benefits, sustained use of, and sustained engagement with virtual care | Shaw et al., 2018 [56] | [56] |
| | Wali et al., 2021 [67] | |
| Sustainability is described as the continued use of virtual visits | Pradhan et al., 2021 [57] | [57] |
| | [57] | [16] |
| | Mohammed et al., 2021 [16] | [61] |
| | Radhakrishnan et aal., 2016 [61] | [65] |
| | Moehr et al., 2006 [65] | [69] |
| | Ware et al., 2018 [69] | |
| Sustainability is described as the continuation of high-volume, high-quality pediatric care | Goldbloom et al., 2022 [58] | [58] |
| Sustainability is defined as, "the ability to continue operating into the future without obvious threats" | Wade et al., 2016 [62] | [62] |
| Sustainability is defined as, "the use of tele-homecare services that hold the promise of being absorbed into routine health care delivery" | Cradduck (2002) [80] | [64] |
| Sustainability is described as a function of the ability of virtual care to integrate with existing workflows | Shah et al., 2020 [70] | [70] |
| Sustainability is described according to the Dynamic Sustainability Framework, whereby the nested relationship between interventions and practice settings defines the term | Chambers et al., 2013 [38] | [71] |
| Sustainability is described as the continued and meaningful use of videoconferencing as an integrated component of routine care | Newbould et al., 2021 [74] | [74] |

## Fidelity

Fidelity was a consideration in describing the sustainability of virtual care interventions. This concept was described as a function of whether the virtual care intervention remained faithful to its intended use (i.e., to what extent did the intervention continue to be used according to its initial design) and its intended outcomes (i.e., to what extent did the virtual care

intervention continue to achieve what it was planned to achieve) [69]. Virtual care interventions included in this scoping review varied according to intended use and intended outcomes; intended uses included clinical care (e.g., telemonitoring, virtual consultations and virtual visits) and health promotion [16,69,70,81] while intended outcomes included specific clinical outcomes, optimized resource use, and improvements in organizational effectiveness/efficiency [58,59,67].

For virtual care interventions with the intended use of clinical care, examples included a virtual chronic disease management program that seeks to connect patients to specialist services through the use of health navigators, and a virtual health coaching and telemonitoring intervention where patients are provided with wireless scales and hemoglobin A1c (HbA1c) test kits to help improve their glycemic control and other risk factors [62]. For interventions with an intended use of chronic disease management, outcomes often focused on specific clinical outcomes (e.g., a target HbA1c level) and organizational efficiency (e.g., cost savings through virtual service delivery). For virtual care interventions with the intended use of health promotion, examples included a health care ambassador program that assists patients with low digital literacy in setting up and engaging in virtual visits [54]. The intended outcomes of health promotion focused virtual care interventions were often to optimize resources (e.g., in contexts of low human resources, using virtual care to connect few community health workers to larger amounts of patients) [60]. Moreover, some virtual care interventions were intended to provide ongoing patient benefits with regard to access and administrative issues related to overall management of their health (e.g., reduce access barriers, alert healthcare providers to changes in health, reduce costs and inefficiencies and costs associated with health service delivery [52,53,56,58,61,64,67].

Fidelity was also determined based on virtual care's ability to be continuously used and achieve expected outcomes, with the length of time outcomes were intended to be sustained varying across studies. Some virtual care interventions were implemented with the intention of being sustained for long-term or ongoing use [61,64–67,69], and others were implemented without any considerations for how long the intervention should be sustained [16,57,58,63,70,71]. Virtual care interventions designed with intentions for ongoing sustainability (e.g., no identified end date) often occurred within the context of continuous resources and specific intended outcomes of the intervention [54,55]. For example, the virtual program MomConnect, a telehealth program focused on maternal care, was developed to be sustained as part of usual care to support a health system that emphasized preventative care. As such, the government ensured commitment to ongoing resources, such as private donor and corporate partnerships, to ensure its sustainability [54].

Other papers that had no expectations or considerations of a timeframe tended to be focused on virtual care use within the context of the coronavirus (COVID-19) pandemic [16,28,82]. In response to the pandemic, limitations on in-person health service delivery were put in place to limit the spread of the virus. These limitations acted as a catalyst for the rapid and widespread uptake of virtual care and many virtual care interventions were implemented without consideration for whether these interventions should or could be continued in the long term [58,59,63]. Many primary care practices introduced or expanded their use of telephone appointments, video visits, virtual triaging, and online appointment booking, but did not specify how long these were to be put in place [16,68]. In these studies, some organizations continued (or planned to continue) the use of these interventions beyond the COVID-19 pandemic [16,82–84]. For example, one review noted that,

> *although health systems scaled virtual health out of necessity, innovating to meet the needs of an immediate crisis is different from sustainable, long-term transformation. Consequently, health systems are beginning to re-evaluate how the virtual health strategies implemented during the COVID-19 surge in virtual health utilization will meet the needs of the system, health care providers, and patients going forward* [71].

## Adaptability

Virtual care interventions in this review were often implemented and intended to be sustained in contexts where things rapidly change; these contexts may include hospital, community, and specialty care settings (e.g., mental health)

[57,64,66]. Many studies (n = 7) in this scoping review noted the importance of examining the context in which virtual care is implemented to assess whether the virtual care intervention was sustained [52,62,63,68,71,76,79]. Since the context in which virtual care interventions are implemented may change, as was seen with COVID-19 where changes in public health restrictions resulted in changes and how often virtual care interventions were used, these studies noted the importance of examining an evolving context and how an organization adapts virtual care interventions to the changing context.

One notable example of how virtual care sustainability related to adaptability to local context is found in the Shaw et al (2018) study where the researchers noted that the evolving conditions within the organization impacted the ability scale-up and sustain video consulting services [56]. In examining the sustainability of video consulting services, determining a patient as having a health condition appropriate condition for the of video consulting and having policies supportive of video consulting alone did not guarantee sustainability. The authors noted that it was the adaptation of the video consulting intervention to the unique financial and organizational contexts of each organization that supported sustainability [56]. In another study, researchers noted that the rapid change to telephone visits for cardiac rehabilitation patients during the COVID-19 pandemic resulted in other attributes of the organization, such as continuous funding for virtual cardiac rehabilitation and coordinating telehealth alongside in-person programming, needing to adapt in order to sustain virtual care intervention [76]. These examples point to the notation that there is reciprocal adaptation in that the virtual care intervention and the environment in which it is being used need to adapt to each other.

In another example, in pre-COVID laparoscopic cholecystectomy (gallbladder removal) surgery, there were typically six in-person interactions with the health system. This changed during the pandemic resulting in the new process for laparoscopic cholecystectomy to be three virtual interactions and only one in-person interaction (the surgery), finding two interactions that could be eliminated [71]. It is suggested in the article that this model will continue to be sustained, which could result in a change in the context of how virtual care is used, leading away from physical interactions, towards an almost equal ratio of physical to virtual interactions [71].

In order to operationalize novel methods of health service delivery, such as in the cholecystectomy example and others, to ensure the sustainability of such initiatives, it is important to detail the intention for the virtual care intervention to be sustained within its context [25,67,69,84,85]. Table 5, below, presents highlighted examples of virtual care interventions that were adapted to their context to facilitate sustainability.

**Other factors related to virtual care sustainability**

Studies in this scoping review pointed to other factors that enabled or hindered the sustainability of virtual care interventions. These included healthcare provider and patient commitment, infrastructure and supports, and the availability of financial and human resources [5,60,62–64,68,70].

**Healthcare provider and patient commitment.** With regards to provider buy-in, some healthcare providers believe that the ongoing use of virtual care interventions is outside their usual scope of and thus sustained use was met with reluctance [60,64]. In one study of provider perspectives on the sustainability of telephone and video visits, patient acceptability was especially important in that healthcare administrated expressed concern of the virtual visits being abandoned if it proved unacceptable to patients [62]. Other findings from the study noted that, "once [patients] are dissatisfied then clinicians go, well, we will go back to what we have always done, and you lose it [telehealth] … if the patients see value then it is more likely that the clinicians will accept it" [62].

**Resources.** Lack of financial resources was noted as a barrier to virtual care sustainability. Healthcare organizations with fewer financial resources reported that upfront costs for equipment and ongoing costs of virtual care use posed a challenge for sustained virtual care use [65,86]. These organizations noted high costs of procuring the necessary telehealth equipment using regular operating budgets and expressed a desire for alternative sources of funding for this upfront cost. Lack of financial resources also prohibited the implementation of training for patients and providers to sustainably use virtual care technology [60,64]. In for-profit healthcare environments, organizations that invested in

**Table 5. Selected examples of adaptability within included studies.**

| Study | Virtual Care Intervention | Context | Evidence of Adaptation |
|---|---|---|---|
| User-Centered Adaptation of an Existing Heart Failure Telemonitoring Program to Ensure Sustainability and Scalability: Qualitative Study [69] | Telemonitoring program for individuals with chronic heart failure (Medly program) | The Medly program is an algorithm-based smartphone App. Patients record daily weight, blood pressure, heart rate, and symptoms. Upon indication of signs of deterioration in a patient's health, the algorithm triggers a self-care message displayed to the patient, an alert is sent to both the patient's care team via automated emails | Enhancements to technical support (e.g., developing a website for patients containing frequently asked questions) was implemented to support feasibility and continued patient engagement. Adjustments to healthcare provider roles and responsibilities related to the intervention (i.e., hiring a registered nurse to take over the primary clinical management of alerts instead and implementing a technical support role – duties that were both formerly performed by the nurse practitioner) |
| "Never waste a good crisis": A qualitative study of the impact of COVID-19 on palliative care in seven hospitals using the Dynamic Sustainability Framework [71] | Virtual palliative care | Seven hospital systems with existing palliative care were funded to expanded their palliative care services over a 3-year period. The expansions focused on: providing care in inpatient, outpatient, and/or community settings; increasing linkages to social services; increasing advance care planning; and supporting caregivers. COVID-19 impacted the expansion by public health mandates to increase hospital capacity by 40%, shelter-in-place orders, supply shortages of personal protective equipment (PPE), and requirement to continue care provision amongst increased COVID-19 cases | For inpatient palliative services, goals of care discussions occurred virtually. To support patient access to technology issues, iPads were procured by the palliative care team to connect patients with families when visitors were not allowed. Provision of iPads also became a valuable way to demonstrate the new palliative care services to other providers. |

infrastructure and equipment noted that an increase in volume was needed to obtain sufficient reimbursements to offset the investments and improve financial performance [54,71].

## Discussion

The objectives of this scoping review were to, 1) develop a conceptualization of sustainability in healthcare settings to contribute to a comprehensive understanding of the optimal implementation and use of virtual care and 2) examine the factors that contribute to virtual care sustainability in healthcare settings. With respect to research objective one, this scoping review conceptualizes the sustainability of virtual care according to its fidelity (i.e., what is being sustained - the extent to which the intervention continues to be used according to its initial design and the extent to which the virtual care intervention continues to achieve what it was planned to achieve) and its adaptability (i.e., how is the intervention sustained - how an organization adapts virtual care interventions to the changing context). With respect to research objective two, this scoping review found that the reciprocal adaptation of technology and organization, patient and healthcare provider commitment to virtual care use and the availability and appropriate use of resources are key contributing factors to sustainability. Findings indicate that there is no "one size fits all" approach to achieving sustainability of virtual care interventions. Important to understanding sustainability of virtual care interventions, is the complexity of the interactions that influence it. Specifically, the factors of fidelity and adaptability are found to be important to understanding the sustainability of virtual care interventions. The findings of this scoping review point to the need to align shifting needs of the context (e.g., changes in organizational priorities) with the functionalities and value (i.e., intended outcomes) of the virtual care intervention as a consideration for sustainability. The purpose of a virtual care intervention, whether for clinical service delivery or health promotion or for long-term or episodic purposes, coupled with the context in which it is used,

provides a roadmap for where, how, and how long the intervention should be sustained. For example, interventions in the health promotion setting may be designed with the purposes of empowering and educating communities and therefore the functionalities of these virtual interventions technologies would be targeted to this purpose; however, this would differ from virtual care intervention implemented in health service delivery settings designed for continuous remote patient monitoring [54,60].

Existing research points to the necessity for studying sustainability as a dynamic process, shifting away from previous conceptualizations of the linear implementation-sustainability pathway [38]. The Dynamic Sustainability Framework (2013) offers one theoretical approach to sustainability which highlights the continuous evolution of an intervention, the context in which it is implemented and the external environment as drivers of sustainability. The Dynamic Sustainability framework emphasizes the ongoing evolution of the intervention, context, and external environment over time. The proposed conceptualization of sustainability presented in this paper align with this theory. In this exploratory scoping review, we highlight fidelity (what about the intervention is being sustained) and adaptability (how the intervention and its environment adapt) as important concepts in understanding the sustainability of virtual care interventions. Exploring the evolution of the aspects of an intervention that should be sustained and how the intervention and its environment change overtime are avenues for future research in this field of study.

The study of sustainability as it relates to virtual healthcare is complex. Current healthcare sustainability research is focused in the areas of identifying barriers and facilitators of sustained services, practices or behaviours [39]; examining contextual factors that contribute to sustainability or lack thereof [37,41,42,44]; and the scale and spread of interventions that have previously been sustained [33,87–89]. This scoping review adds to this field of study by highlighting the context of the COVID-19 pandemic as a driver of creativity and innovation, whereby virtual care interventions had to be rapidly sustained. The pandemic challenged the healthcare system and had a transformational impact on health service delivery [16]. The COVID-19 pandemic was disruptive to the healthcare system, influencing resources, infrastructure and funding policy, and funding — components of the external environment that challenged and tested the fidelity of both existing and newly sustained virtual care interventions. Because the implementation and impact of interventions cannot be divorced from the unique contexts in which they are implemented, it is important to understand the dynamic fit between virtual care interventions and how they adapted to the environment created by the pandemic [71]. The literature of disruptive and sustaining innovations may help to provide some insight to the sustainability of virtual care interventions following the pandemic. Authors in this field of work argue that virtual care can be sustained through capitalizing on both sustaining innovation (i.e., incremental improvement upon what was already done for patients pre-pandemic) and through disruptive innovation (i.e., novel solutions for patients whose need are not currently being served) [90]. Scholars in this field argue that virtual care can be sustained through both sustaining innovations (i.e., incremental improvements to existing care delivery models) and disruptive innovations (i.e., novel solutions addressing unmet needs of underserved patient populations) (1, 2). However, it is important to recognize that many healthcare systems adopted virtual care not just as an innovative approach, but as a necessary response to deep-seated structural challenges in healthcare delivery (3). In resource-limited settings where transportation to healthcare facilities remains problematic and specialist providers are scarce, virtual care often represents the only feasible option for timely medical attention rather than merely being an innovative alternative. However, the long-term sustainability of these solutions depends on addressing fundamental feasibility challenges, including reliable internet connectivity, digital literacy gaps among providers and patients, and the organizational capacity to integrate technologies into clinical workflows (4–7).

The pandemic quickly demanded that patients, health care providers, and health systems adapt virtual care at an unprecedented pace of change. Continuing to manage this rate of change will remain a challenge post-pandemic and maintaining these elevated levels of virtual health adoption and sustained use across health systems will require future inquiry [91]. Beyond this line of inquiry, analyses of how the sustainability of virtual care interventions can encourage or discourage the appropriate use of services should also be conducted.; the literature notes that the optimal applications of

virtual care interventions would support and enable the provision of appropriate level and intensity of care at the point of need by appropriate providers in an appropriate setting [79,86].

While the balance between fidelity and adaptability has been examined by scholars in the broader field of sustainability [50], this topic should be examined more closely as it relates to virtual care sustainability. Questions about what aspects of an intervention should or should not be sustained, and how to maximize the ability of an intervention to adapt to the environment as vice versa remain. A question for further discussion that emerged from this review is when thinking about this concept, is: what are the feasible expectations for a virtual care intervention to evolve within environmental and organization contexts while attaining its intended objectives? As the requirements to continue these outcomes shifted, so may the need to sustain the virtual care intervention.

## Limitations

While this scoping review contributed to establishing a comprehensive conceptualization of virtual care sustainability, there are some limitations to consider. First, we did not limit this review to a specific type of virtual care intervention or to a specific healthcare setting, thereby preventing an in-depth examination of sustainability factors unique to a typology or setting of virtual care. Our broad perspective may omit nuances specific to the sustainability factors unique to specific virtual care interventions. Secondly, as this scoping review was exploratory in nature, we were unable to develop a comprehensive definition of sustainability for virtual care interventions. Additionally, due to the large number of papers focused on rapid change environment of the COVID-19, some findings may only be relevant to this context. The proposed conceptualization of sustainability in this paper is highly attuned to rapid change environments where adaptability might be more important, whereas in more stable environments might be able to focus more on fidelity. This is an area for future research. This scoping review establishes a starting point for which other empirical and theoretical studies can build upon to gain insight to factors of sustainability in different healthcare contexts and to establish a universal definition of virtual care sustainability.

## Conclusions

This research paper strives to contribute to the ongoing discourse surrounding virtual care by providing a comprehensive understanding of its sustainability. By proposing a conceptualization of sustainability as it specifically relates to virtual care intervention and summarizing factors that contribute to virtual care sustainability, we have illuminated the interplay of factors shaping the longevity and effectiveness of virtual care in healthcare settings. The global outbreak of COVID-19 has not only accelerated the adoption of digital solutions but has also underscored the indispensable role of virtual care in healthcare resilience. As the pandemic continues to reshape conventional healthcare delivery, the sustainability of virtual care emerges as a pivotal consideration in future health service delivery.

## Author contributions

**Conceptualization:** Tujuanna Austin, Ross Baker, Carolyn Steele Gray.

**Data curation:** Tujuanna Austin.

**Formal analysis:** Tujuanna Austin, Farah Tahsin.

**Investigation:** Tujuanna Austin.

**Methodology:** Tujuanna Austin, Farah Tahsin.

**Project administration:** Tujuanna Austin.

**Supervision:** Darren Larsen, Ross Baker, Carolyn Steele Gray.

**Validation:** Tujuanna Austin, Farah Tahsin.

**Writing – original draft:** Tujuanna Austin.

**Writing – review & editing:** Tujuanna Austin, Farah Tahsin, Darren Larsen, Ross Baker, Carolyn Steele Gray.

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
