## [Decision Letter · Decision Letter 0]

PDIG-D-24-00462Exploring the sustainability of virtual care interventions: A scoping reviewPLOS Digital Health Dear Dr. Austin, Thank you for submitting your manuscript to PLOS Digital Health. After careful consideration, we feel that it has merit but does not fully meet PLOS Digital Health's publication criteria as it currently stands. Therefore, we invite you to submit a revised version of the manuscript that addresses the points raised during the review process. Please submit your revised manuscript within 60 days Apr 11 2025 11:59PM. If you will need more time than this to complete your revisions, please reply to this message or contact the journal office at digitalhealth@plos.org. Please include the following items when submitting your revised manuscript:* A rebuttal letter that responds to each point raised by the editor and reviewer(s). You should upload this letter as a separate file labeled 'Response to Reviewers '. This file does not need to include responses to any formatting updates and technical items listed in the 'Journal Requirements' section below.* A marked-up copy of your manuscript that highlights changes made to the original version. You should upload this as a separate file labeled 'Revised Manuscript with Track Changes '.* An unmarked version of your revised paper without tracked changes. You should upload this as a separate file labeled 'Manuscript '. If you would like to make changes to your financial disclosure, competing interests statement, or data availability statement, please make these updates within the submission form at the time of resubmission. Guidelines for resubmitting your figure files are available below the reviewer comments at the end of this letter. We look forward to receiving your revised manuscript. Kind regards, Simon ReifAcademic EditorPLOS Digital Health Simon ReifAcademic EditorPLOS Digital Health Leo Anthony CeliEditor-in-ChiefPLOS Digital Healthorcid.org/0000-0001-6712-6626 **Journal Requirements:**

1. We ask that a manuscript source file is provided at Revision. Please upload your manuscript file as a .doc, .docx, .rtf or .tex.

2. Please provide separate figure files in .tif or .eps format.

3. In the online submission form, you indicated that "The data that support the findings of this study are available from the corresponding author, TA, upon reasonable request.". 

a. In a public repository, 

b. Within the manuscript itself, or 

c. Uploaded as supplementary information.

 **Additional Editor Comments (if provided):** Both reviewers provide very sound suggestions on how the paper needs to get improved prior to publication. In addition, please rewrite the introduction to be less Canada focused since the review is not restricted to a specific country.**Reviewers' Comments:** Reviewer's Responses to Questions

**Comments to the Author**

1. Does this manuscript meet PLOS Digital Health’s publication criteria ? Is the manuscript technically sound, and do the data support the conclusions? The manuscript must describe methodologically and ethically rigorous research with conclusions that are appropriately drawn based on the data presented.

Reviewer #1: Partly

Reviewer #2: Yes

2. Has the statistical analysis been performed appropriately and rigorously?

Reviewer #1: N/A

Reviewer #2: N/A

3. Have the authors made all data underlying the findings in their manuscript fully available (please refer to the Data Availability Statement at the start of the manuscript PDF file)?

Reviewer #1: Yes

Reviewer #2: Yes

4. Is the manuscript presented in an intelligible fashion and written in standard English?

Reviewer #1: Yes

Reviewer #2: Yes

5. Review Comments to the Author

Reviewer #1: The paper provides a scoping review of virtual care sustainability thus the continued use of virtual care. While the object of sustainability with respect to virtual care provides a unique research area, I fear that the given paper’s final list of studies may need more additions.

1. Materials and methods

Many words are often used to describe different or same things related to virtual care. Regarding the additional provided document “Search Strings”, I fear that the authors have lost many valuable papers that may use for example “video consultation/consulting” or “video appointments” instead of “virtual care”. Adding “remote” or “video” might provide more papers to use. In the same vein, as the paper mentioned as well, the concept of sustainability may be part of a paper without using the search terms that appear in the search term document. Generally, 21 papers seems to be too few papers in the given context, hence my ideas to improve the search strategy.

I am also confused by the stated research question on page 6. To find “what is known” about sustainability in virtual care does only very broadly match the mentioned goals of a) providing a conceptual framework and b) finding factors that influence sustainability. In the same vein, I became more confused when reading the paper on whether the authors want to keep a narrow or wider view on sustainability. Their exclusion criteria regarding sustainability may limit potential candidates too much. As a reader “do not conceptualize/describe/define sustainability” is not entirely clear to me how much gets lost.

2. COVID

At the end of the paper, page 30 line 379-380, the authors mention the study’s addition with respect to covid in the setting. I fear however that the paper is not clear enough on Covid in general. The conclusion of covid being the driver for innovation and creativity does not match the overall paper’s mentioning of covid and I was surprised to read that sentence in the conclusion. The paper may benefit from more clear explanations regarding this inside the main text. For now, I am confused on how the paper tries to argue with regards to covid.

3. Fidelity and adaptability

To my understanding Fidelity and adaptability open the paper section on how factors influence sustainability of virtual care. To the reader this section could benefit from more structure and introduction. After the conceptual framework, a section of influencing factors marks the second big section in this paper according to the abstract. More introduction and general larger frame for this chapter may help convey the findings from the scoping review.

Minor comments

a) Typo in flow chart “conceputalize”

b) Page 3 line 71 to 74 sentence structure confusing

c) Page 3 line 75 hospitals s missing

d) Page 4 line 94 how can non-essential surgeries backlog be solved by virtual care needs more explanation

e) Page 8 line 170 “kappa score” not clear to me

f) Page 23 line 239 “paired” not clear

g) Page 23 line 253 and 254 confusing sentence

h) Page 25 line 290-292 confusing sentences

i) Page 29 line 350 does not match previous sentence part

j) Page 31 line 405 and 406 two times “question”

Reviewer #2: This scoping review charted the current evidence reporting on virtual care interventions in a healthcare setting. It follows appropriate scoping review guidelines an overview of key factors to consider in sustainable virtual care through the developed conceptual framework. More detail should be provided on the rationale for structuring the conceptual framework. Only a few sentences were provided for this where it is a central part of the review. Specific comments are provided below.

Note – “L” = line in manuscript

L34-54: the conceptual framework could be more clearly outlined in the results section of the abstract.

L56-68: consider briefly defining what a conceptual framework is and why it is used in the author summary for the benefit of non-experts.

L124-129: build a stronger argument for the factors considered in the conceptual framework. Why are fidelity and adaptability considered to be important? Why does it make sense to develop a conceptual framework focus only on these two factors?

L139-141: consider linking the research question to the review objectives (e.g., in asking this question, a conceptual framework will be developed, etc.)

L154-156: these criteria could be made clearly – particularly this part: “…or reference to the study authors’ provided 156 definition or conceptualization (including contributing factors to) of the term sustainability”.

L168-172: state how many reviewers extracted data.

L181-190: how was the data summarised to get a descriptive understanding and develop patterns? What kind of software/method was used to organise data for analysis? If an established method was used, state this. How many team members were involved in this process? Etc.

L390-392: The term “disruptive innovation” is used here which represents the general focus of this review which is on the novel application of technology to virtual care. To counterbalance this theme, I suggest including a paragraph that acknowledges structural factors at play that force the use of such technologies (e.g., understaffed facilities, poor transport to care facilities, etc.). It could link in with the point made in the introduction regarding when it is and is not appropriate to use such technologies. I see that the feasibility of these technologies is raised as a question for future researchers to answer to this point could also link in with that.

6. PLOS authors have the option to publish the peer review history of their article (what does this mean? ). If published, this will include your full peer review and any attached files.

**Do you want your identity to be public for this peer review?** For information about this choice, including consent withdrawal, please see our Privacy Policy .

Reviewer #1: No

Reviewer #2: **Yes: ** David Healy

---

## [Editor Report · Decision Letter 1]

Exploring the sustainability of virtual care interventions: A scoping review

PDIG-D-24-00462R1

Dear Ms. Austin,

We are pleased to inform you that your manuscript 'Exploring the sustainability of virtual care interventions: A scoping review' has been provisionally accepted for publication in PLOS Digital Health.

Best regards,

Simon Reif

Academic Editor

PLOS Digital Health